# Evaluation of ocular biometry in primary angle-closure disease with two swept source optical coherence tomography devices

**Boonsong Wanichwecharungruang**[1,2], **Anyarak Amornpetchsathaporn**[1], **Wisakorn Wongwijitsook**[1], **Kittipong Kongsomboon**[3], **Somporn Chantra**[1]*

1 Department of Ophthalmology, Rajavithi Hospital and Rangsit Medical College, Bangkok, Thailand,
2 Department of Ophthalmology, Priest Hospital, Bangkok, Thailand, 3 Department of Preventive and Social Medicine, Faculty of Medicine, Srinakharinwirot University, Nakhon Nayok, Thailand

* chantrasomporn@yahoo.com

## Abstract

### Purpose

To investigate agreement between 2 swept source OCT biometers, IOL Master700 and Anterion, in various ocular biometry and intraocular lens (IOL) calculations of primary angle-closure disease (PACD).

### Setting

Rajavithi Hospital, Bangkok, Thailand.

### Design

Prospective comparative study.

### Methods

This study conducted in a tertiary eye care center involving biometric measurements obtained with 2 devices in phakic eye with diagnosis of PACD. Mean difference and intra-class correlation coefficient (ICC) with confidence limits were assessed, and calculations of estimated residual refraction of the IOL were analysed using Barrett's formula.

### Results

Sixty-nine eyes from 45 PACD patients were enrolled for the study. Excellent agreement of various parameters was revealed, with ICC (confidence limits) of K1 = 0.953 (0.861–0.979), K2 = 0.950 (0.778–0.98), ACD = 0.932 (0.529–0.978), WTW = 0.775 (0.477–0.888), and LT = 0.947 (0.905–0.97). Mean difference of axial length (AL) was -0.01 ± 0.02 mm with ICC of 1.000. IOL calculation was assessed with Barrett's formula, and Bland-Altman plot showed excellent agreement in the results of the 2 devices for the IOL power and estimated post-operative residual refraction (EPR).

**Data Availability Statement:** All relevant data are within the paper and its Supporting information files.

**Funding:** SC received fund from Rajavithi research funds with grant number 241/2563. The funders had no role in study design, data collection and analysis, decision to publish, or preparation of the manuscript.

**Competing interests:** The authors have declared that no competing interests exist.

**Abbreviations:** ACD, Anterior chamber depth; AL, Axial length; CCT, Central corneal thickness; EPR, Estimated post-operative residual refraction; ICC, Intraclass correlation coefficient; K, Keratometry; LAF, Lens-axial length factor; LT, Lens thickness; PAC, Primary angle closure; PACD, Primary angle closure disease; PACG, Primary angle closure glaucoma; PACS, Primary angle closure suspect; WTW, White-to-white corneal diameter.

## Conclusions

Mean differences of biometric parameters, obtained with IOL Master700 and Anterion, were small, and ICC showed excellent concordance. No clinical relevance in calculation of IOL power was found, and the two devices appeared to be comparably effective in clinical practice.

## Introduction

Primary angle-closure glaucoma (PACG) is one of the leading causes of irreversible blindness worldwide. Its global prevalence is showing an upward trend, and it is expected to reach about 66.8 million cases by 2040. Its prevalence is highest in Asia, and people from this regiohave the greatest number of cases of blindness resulting from glaucoma [1]. Primary angle-closure disease (PACD) is a spectrum of this eye condition, including primary angle-closure suspect (PACS), primary angle-closure (PAC) and PACG. PACD is an anatomic disorder, typically of relatively small eye with shallow anterior chamber depth (ACD), thick lens, shorter axial length (AL), small corneal curvature, and relatively anterior lens position. Pupillary block, plateau iris and phacomorphic mechanisms play major roles in the pathophysiology of the disease, and pupillary block is thought to be the most common causal mechanism. Laser peripheral iridotomy (LPI) alleviates the effects of pupillary block; however, in the LIWAN study, 58% of patients were found to still have apposition irido-corneal angle [2]. Irido-corneal angle can be closed by a crowded anterior chamber, leading to intraocular pressure (IOP) elevation and glaucomatous optic neuropathy (GON) resulting in visual field loss and irreversible blindness in these patients. Visual field loss in PACD appears to be worse than in primary open-angle glaucoma (POAG) [3].

For decades, phacoemulsification and intraocular lens implantation (PEI) has been proposed for initial management of PACD [4, 5]. PEI relieves pupillary block, deepening ACD and widening irido-corneal angle [6], while at the same time preserving the conjunctiva for future trabeculectomy if needed.

Biometry is essential in evaluating ocular dimensions and calculating IOL power in PACD, and ocular biometers have been evolving in recent years. Optical coherence tomography (OCT) includes partial coherence interferometry (PCI), and swept source OCT (SS-OCT) is in general use in clinical practice. The agreement of the calculations of these devices has been studied by many investigators; however, research into their agreement in PACD patients has been sparse so far. A newer SS-OCT, Anterion, is now available and its performance needs to be evaluated.

## Materials and methods

A prospective comparative study of ocular biometry of 2 devices, IOLMaster 700 (Carl Zeiss Meditec, CA, USA) and Anterion (Heidelberg Engineering GmbH, Germany), for use with PACD patients, was conducted in a referral eye center, in Rajavithi Hospital, Ministry of Public Health of Thailand. The research protocol was approved by the Ethics Committee Rajavithi Hospital in April 2020, and the study was performed between May and December 2020 following all of the guidelines for experimental investigation in human subjects required by the Ethics Committee. All investigations were carried out in accordance with the Declaration of Helsinki, and informed consent forms were read and signed by all participants.

### Inclusion and exclusion criteria

We enrolled participants from the glaucoma clinic at Rajavithi Hospital, Bangkok, Thailand. Inclusion criteria were age $\geq$ 40 years old and phakic eye with diagnosis of PACD, including

PACS, PAC, and PACG as classified by Foster et al [7]. PACS was defined as Shaffer's gonioscopic grading $\leq 2$ in at least 2 quadrants, with normal IOP and without glaucomatous optic disc and visual field defect; PAC was classified as PACS with IOP $> 21$ mmHg, presence of PAS and without any glaucomatous changes; and PACG was defined as PAC with glaucomatous changes. Patients who had glaucoma medication or previous laser treatment (iridotomy, iridoplasty, trabeculoplasty) were eligible. Exclusion criteria were patients who had opaque optical media, dense cataract, anterior and/or posterior segment diseases such as advanced pterygium, diabetic retinopathy, or maculopathy. Participants who had previous history of ocular trauma, ocular surgery, e.g., refractive surgery, cataract, glaucoma (trabeculectomy and/or glaucoma tube shunt), and vitreoretinal surgery were also excluded.

## Examinations

Ocular examination including Snellen visual acuity with logMar conversion, autorefraction (RC-5000, Tomey, Japan), slit lamp, Goldman applanation tonometry, cup/disc (C/D) and fundus ophthalmoscopy were evaluated. Dynamic gonioscopy was performed for all patients by a senior glaucoma specialist (BW).

## Swept source OCT devices

Swept source OCT has been an emerging technology for IOL calculation for a number of years. SS-OCT applies a tunable light source to a beam splitter, diverting one arm of the scanning beam to the ocular structure and the other to a reference mirror. Both of the coherence beams reflect to the point detector, assessing Fourier transformation to calculate ocular dimensions [8]. IOL Master700 applies a 1050 nm light source with 2,000 A scan/sec, capturing scan depth of 44 mm and axial resolution of 22 microns, while Anterion applies a 1,300 nm light source with 50,000 A scan/sec, capturing 32 mm scan depth, 16.5 mm scan width, axial resolution $< 10$ microns and lateral resolution of 30–45 microns [8].

The 2 devices, IOLMaster700 and Anterion, were scanned in random sequences by trained technicians within the same visit in a standard illuminated room. No pupillary dilation was required. We checked all image quality and segmentation to ensure that the devices were correctly marked. Biometric parameters of ACD, AL, LT, K1, K2, astigmatism, and white to white corneal diameter (WTW) were recorded in Excel and transferred to SPSS (version 20, SPSS inc., IBM, Chicago, USA) for analyses.

## Sample size estimation

Sample size calculation was performed using the formula for estimating correlation coefficient with a type I error ($\alpha$) of 0.05 and type II error ($\beta$) of 0.2. The authors estimated the correlation of the devices, using the correlation coefficient (r) of 0.7 for the sample size calculation.

The standard normal deviate for $\alpha = Z_\alpha = 1.9600$

The standard normal deviate for $\beta = Z_\beta = 0.8416$

$$C = 0.5 * ln[(1 + r)/(1 - r)] = 0.8673$$

$$\text{Total sample size} = N = [(Z_\alpha + Z_\beta)/C]^2 + 3 = 13$$

The output indicated that at least 13 subjects should be enrolled. And, in the anticipation of a dropout rate of 30%, the minimum sample size required was 17.

## Outcome measurement and statistical analyses

We tested normal distribution of data with Kolmogorov-Smirnov test. Comparison of the biometric parameters were analyzed with independent t-test for parametric datasets, and Mann-Whitney U test was used for non-parametric datasets. Statistical significance was set at $p < 0.05$.

The main outcome measured was agreement of the biometric parameters of the 2 devices. Intraclass correlation coefficient (ICC) and confidence limits were analyzed. ICC was classified as follows: < 0.4, poor; 0.4–0.59, fair; 0.60–0.74, good; and 0.75–1.00, excellent agreement [9].

Subgroup analyses of PACD patients who had ACD < 2.4 mm and > 2.4 mm were performed, and agreement of IOL calculation by the 2 devices was also analyzed. We assessed estimated IOL power and estimated residual post-operative refraction with Barrett Universal II online formula (https://calc.apacrs.org/barrett_universal 2105, accessed July 15, 2021). The formula required AL, K1, K2, and ACD, with option of LT and WTW. We used an IOL model SN60WF (Alcon Laboratories, Fort Worth, TX, USA) with A constant 119.0 for the formula. Bland-Altman plot was constructed for the IOL power and the EPR.

## Results

We recruited 50 patients for this research. Two patients were excluded because of dense cataract and failure to be scanned with both devices, and a further three patients were excluded because they had epiretinal membrane (2 patients) and severe dry eye (1 patient). Seventy PACD eyes from 45 patients, 16 males and 29 females, were therefore enrolled in the study. Anterion failed to obtain AL in one eye; therefore, 69 eyes were analyzed. Mean age, visual acuity, spherical equivalent, IOP and cup-to-disc ratio are displayed in Table 1.

Ocular biometry of PACD demonstrated that each biometry carried out, except the EPR of IOL model SN60WF, showed significant differences between the 2 devices (Table 2). Mean difference was small in AL, at—0.01 ± 0.02 mm, and EPR, at 0.01 ± 0.17 D. Intraclass correlation coefficient of all parameters demonstrated excellent agreement between the 2 devices, and ICC of AL was 1.000:

**Table 1. Number of PACD patients and their demographic data.**

| Data | PACD |
|---|---|
| Number of patients | 45 |
| • Male | 16 (35.6%) |
| • Female | 29 (64.4%) |
| Number of eyes | 69 |
| Age (years) | 63.27±8.01 (48–77) |
| Diagnosis, N eye (%) | |
| • PACS | 45 (65.22%) |
| • PAC | 11 (15.94%) |
| • PACG | 13 (18.84%) |
| Visual acuity (logMAR) | 0.32±0.48 (0.0–2.7) |
| Spherical equivalent (D) | 0.45±1.97 (-4.0–5.5) |
| Intraocular pressure (mmHg) | 16.54±4.27 (6–34) |
| Cup-to-disc ratio | 0.62±0.19 (0.3–1.0) |

PACD: Primary angle closure disease, PACS: Primary angle closure suspect, PAC: Primary angle closure
PACG: Primary angle closure glaucoma, D: diopter(s)

**Table 2. Comparison of various biometry and intraclass correlation coefficient between IOLMaster700 and Anterion in PACD.**

| Parameter | IOLMaster 700 | Anterion | Mean difference ±SD | P value | ICC | Confident limit | |
|---|---|---|---|---|---|---|---|
| | Mean ± SD | Mean ± SD | | | | lower | upper |
| ACD (mm) | 2.48 ± 0.37 | 2.58 ± 0.33 | -0.10 ± 0.09 | <0.001[a*] | 0.932 | 0.529 | 0.978 |
| AL (mm) | 22.94 ± 0.83 | 22.95 ± 0.83 | -0.01 ± 0.02 | 0.003[a*] | 1.000 | 1 | 1 |
| K1 (D) | 43.66 ± 1.22 | 43.47 ± 1.18 | 0.20 ± 0.29 | <0.001[a*] | 0.953 | 0.861 | 0.979 |
| K2 (D) | 44.62 ± 1.37 | 44.37 ± 1.28 | 0.25 ± 0.27 | <0.001[a*] | 0.950 | 0.778 | 0.98 |
| LT (mm) | 5.00 ± 0.32 | 5.04 ± 0.30 | -0.04 ± 0.10 | 0.003[a*] | 0.947 | 0.905 | 0.97 |
| WTW (mm) | 11.71 ± 0.44 | 11.52 ± 0.51 | 0.19 ± 0.28 | <0.001[a*] | 0.775 | 0.477 | 0.888 |
| LAF | 2.18 ± 0.17 | 2.20 ± 0.17 | -0.02 ± 0.04 | 0.005[a*] | 0.967 | 0.941 | 0.981 |
| IOL power (D) | 22.30 ± 2.23 | 22.64 ± 2.34 | -0.34 ± 0.34 | <0.001[b*] | 0.978 | 0.831 | 0.993 |
| EPR (D) | 0.00 ± 0.11 | -0.02 ± 0.11 | 0.01 ± 0.17 | 0.488[b] | -0.249 | -0.461 | -0.11 |

Values are presented as mean ± SD,

* = significant P value <0.05

[a] p-value by independent t-test

[b] p-value by Mann-Whitney U test

PACD: Primary angle closure disease, ACD: Anterior chamber depth, AL: Axial length, K: Keratometry, LT: Lens thickness, WTW: White-to-white corneal diameter,

LAF: Lens-axial length factor, EPR: Estimated post-operative residual refraction, ICC: intraclass correlation coefficient, IOL: intraocular lens, D: diopter(s)

## Subgroup analysis in PACD

We performed subgroup analysis at cut off ACD of 2.4 mm, obtained with the IOLMaster700. From the 69 eyes, we had 24 with ACD ≤ 2.4 mm and 45 with ACD > 2.4 mm.

**Subgroup ACD ≤ 2.4 mm.** Ocular biometry of the eyes with ACD equal to or less than 2.4 mm measured by IOLMaster 700 demonstrated that all biometries, except of the LT, LAF and EPR, were significantly different between the 2 devices. Mean difference of EPR was 0 ± 0.11 D. Mean difference of AL was close to zero and ICC was 1.000. The other parameters demonstrated excellent agreement between the 2 devices: (Table 3).

**Subgroup with ACD > 2.4 mm.** Ocular biometry of eyes with ACD of more than 2.4 mm measured by IOLMaster 700 demonstrated that all biometries, except the EPR of IOL model SN60WF, were significantly different between the 2 devices. All parameters measured by the two devices demonstrated excellent agreement (Table 4).

## Intraocular lens calculation and estimation of residual post-operative refraction

We applied the required biometric parameters of both devices to Barrett Universal II formula for IOL calculation. The IOL power differences between the devices' measurements are shown with Bland-Altman plot (Figs 1A–1C and 2A–2C):

**Descriptive data of IOL calculation.** IOL power, calculated by IOLMaster700 and Anterion, was shown as:

- Same IOL power: 24 eyes (34.8%)

- ±0.5 D: 39 eyes (56.5%)

- ±1 D: 6 eyes (8.7%)

**Table 3. Comparison of various biometry and intraclass correlation coefficient in PACD with ACD ≤ 2.4 mm, between IOLMaster700 and Anterion.**

| Parameter | IOLMaster 700 | Anterion | Mean difference ±SD | P value | ICC | Confident limit | |
|---|---|---|---|---|---|---|---|
| | Mean ± SD | Mean ± SD | | | | lower | upper |
| ACD (mm) | 2.13 ± 0.27 | 2.25 ± 0.18 | -0.13 ± 0.14 | <0.001[b*] | 0.701 | 0.176 | 0.885 |
| AL (mm) | 22.75 ± 0.80 | 22.76 ± 0.80 | -0.01 ± 0.02 | 0.017[a*] | 1.000 | 0.999 | 1 |
| K1 (D) | 43.78 ± 1.38 | 43.56 ± 1.34 | 0.22 ± 0.25 | <0.001[a*] | 0.971 | 0.824 | 0.991 |
| K2 (D) | 44.80 ± 1.68 | 44.53 ± 1.61 | 0.27 ± 0.23 | <0.001[a*] | 0.978 | 0.712 | 0.994 |
| LT (mm) | 5.15 ± 0.23 | 5.17 ± 0.21 | -0.02 ± 0.14 | 0.583[a] | 0.819 | 0.626 | 0.917 |
| WTW (mm) | 11.54 ± 0.50 | 11.31 ± 0.60 | 0.23 ± 0.33 | <0.001[b*] | 0.764 | 0.399 | 0.903 |
| LAF | 2.27 ± 0.11 | 2.28 ± 0.11 | -0.00 ± 0.06 | 0.706[a] | 0.862 | 0.703 | 0.939 |
| IOL power (D) | 22.60 ± 1.55 | 23.00 ± 1.68 | -0.40 ± 0.36 | <0.001[a*] | 0.948 | 0.556 | 0.985 |
| EPR (D) | 0.00 ± 0.11 | -0.02 ± 0.13 | 0.02 ± 0.19 | 0.584[b] | -0.269 | -0.627 | 0.159 |

Values are presented as mean ± SD

* = significant P value <0.05

[a] p-value by independent t-test,

[b] p-value by Mann-Whitney U test

PACD: Primary angle closure disease, ACD: Anterior chamber depth, AL: Axial length, K: Keratometry, LT: Lens thickness, WTW: White-to-white corneal diameter,

LAF: Lens-axial length factor, EPR: Estimated post-operative residual refraction, ICC: intraclass correlation coefficient, IOL: intraocular lens, D: diopter(s)

## Discussion

The present study demonstrated excellent agreement between the 2 SS-OCT devices in biometric measurement of PACD and its subgroups. Mean difference of AL in the present study was close to zero, -0.01 ± 0.02 mm, and ICC of AL was 1.000. Since, AL is a key factor for IOL calculation, Bland-Altman plot of the estimated residual refraction with Barrett formula showed that no case was outside the 95% LoA. Excluding dense cataract, both devices had high rates of successful scanning. Only 1 of the eye (1.4%) measurements of AL by Anterion failed; therefore, they appeared to be comparably useful in IOL calculation.

**Table 4. Comparison of various biometry and intraclass correlation coefficient in PACD with ACD >2.4 mm, between IOLMaster700 and Anterion.**

| Parameter | IOLMaster 700 | Anterion | Mean difference ±SD | P value | ICC | Confident limit | |
|---|---|---|---|---|---|---|---|
| | Mean ± SD | Mean ± SD | | | | lower | upper |
| ACD (mm) | 2.68 ± 0.25 | 2.77 ± 0.24 | -0.08 ± 0.02 | <0.001[b*] | 0.945 | 0.007 | 0.989 |
| AL (mm) | 23.07 ± 0.85 | 23.07 ± 0.85 | -0.01 ± 0.02 | 0.01[a] | 1.000 | 1 | 1 |
| K1 (D) | 43.64 ± 1.14 | 43.45 ± 1.11 | 0.19 ± 0.27 | <0.001[b*] | 0.958 | 0.859 | 0.982 |
| K2 (D) | 44.57 ± 1.17 | 44.32 ± 1.07 | 0.25 ± 0.29 | <0.001[a*] | 0.944 | 0.717 | 0.98 |
| LT (mm) | 4.92 ± 0.34 | 4.97 ± 0.32 | -0.05 ± 0.06 | <0.001[a*] | 0.973 | 0.864 | 0.99 |
| WTW (mm) | 11.79 ± 0.38 | 11.62 ± 0.43 | 0.17 ± 0.25 | <0.001[b*] | 0.753 | 0.423 | 0.884 |
| LAF | 2.14 ± 0.18 | 2.18 ± 0.18 | -0.02 ± 0.02 | <0.001[a*] | 0.984 | 0.916 | 0.994 |
| IOL power (D) | 22.03 ± 2.53 | 22.35 ± 2.64 | -0.31 ± 0.33 | <0.001[b*] | 0.985 | 0.889 | 0.995 |
| EPR (D) | 0.00 ± 0.10 | -0.01 ± 0.10 | 0.01 ± 0.16 | 0.718[b] | -0.226 | -0.501 | 0.084 |

Values are presented as mean ± SD

* = significant P value <0.05

[a] p-value by independent t-test

[b] p-value by Mann-Whitney U test

PACD: Primary angle closure disease, ACD: Anterior chamber depth, AL: Axial length, K: Keratometry, LT: Lens thickness, WTW: White-to-white corneal diameter,

LAF: Lens-axial length factor, EPR: Estimated post-operative residual refraction, ICC: intraclass correlation coefficient, IOL: intraocular lens, D: diopter(s)

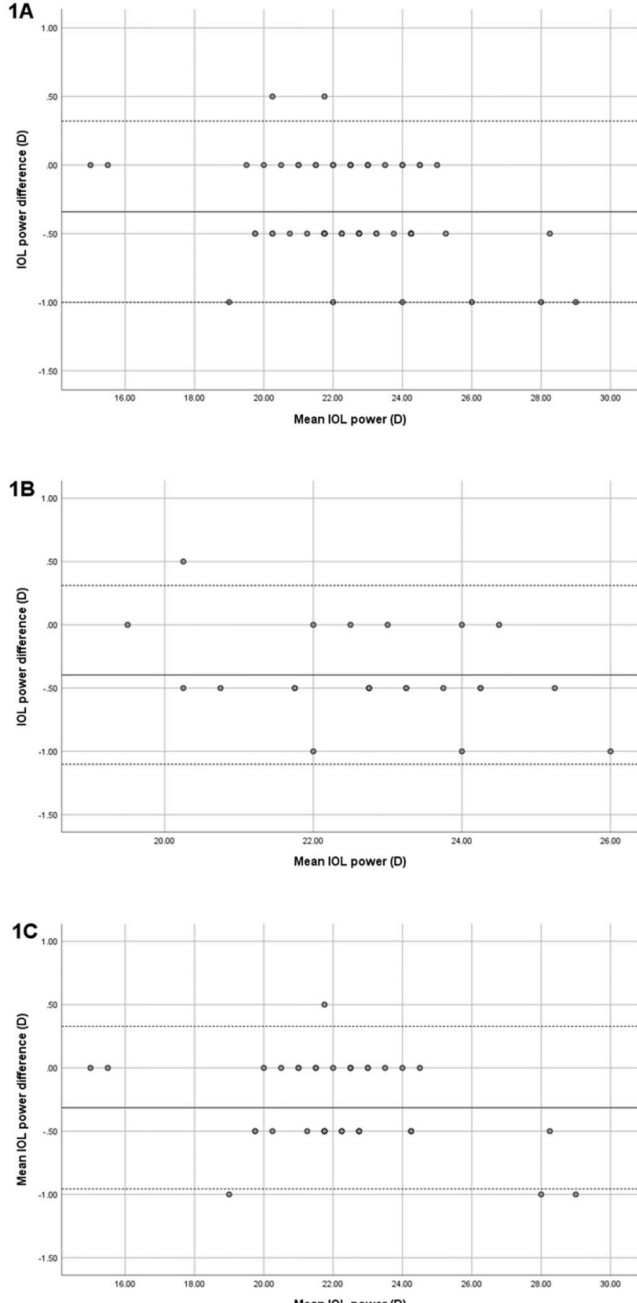

**Fig 1. The Bland-Altman plot of the IOL power using parameters from IOLMaster700 and Anterion.** The solid line represents the mean difference, whereas dotted lines on both sides represent the upper and lower 95% LoA. **A:** In all PACD eyes, The plot demonstrates that only two eyes from total 69 eyes were out of the 95% LoAs, indicating excellent agreement in IOL power. **B:** In the eyes with ACD≤2.4 mm, The plot demonstrates that only one eye from 24 eyes were out of the 95% LoAs, indicating excellent agreement in IOL power. **C:** In the eyes with ACD>2.4 mm. The plot demonstrates that only one eye from 45 eyes were out of the 95% LoAs, indicating excellent agreement in IOL power.

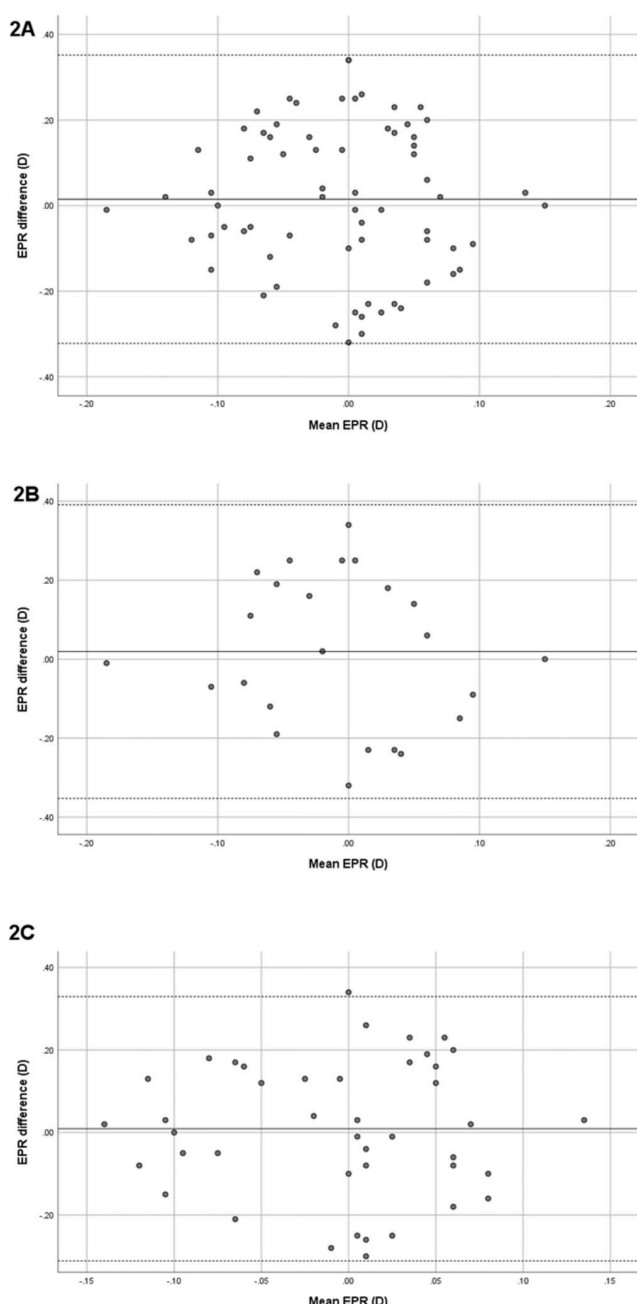

**Fig 2. The EPR between the devices' measurements.** The Bland-Altman plot of the EPR using parameters from IOLMaster700 and Anterion of all PACD eyes. The solid line represents the mean difference, whereas dotted lines on both sides represent the upper and lower 95% LoA. **A:** In all PACD eyes, the plot demonstrates that all cases were within the 95% LoAs, indicating excellent agreement in EPR. **B:** In the eyes with ACD ≤ 2.4 mm, all cases were within the 95% LoAs, which indicates excellent agreement between the devices. **C:** In the eyes with ACD > 2.4 mm, the plot demonstrates that only one eye was out of the 95% LoAs, indicating excellent agreement between the devices.

Optical biometry has been widely used in ophthalmic clinics for many years, constituting non-contact, non-invasive systems for IOL calculation. IOL Master500 (Carl Zeiss Meditec, CA, USA), a partial coherence interferometry (PCI) biometer, was one of the pioneers in the field. A newer platform of SS-OCT, namely IOL Master700, demonstrates good

agreement with IOL Master500 [10], and it is being adopted as a substitute for its predecessor.

Anterion, a recent SS-OCT, has been studied for intra-device repeatability of biometric measurement. Ruiz-Mesa et al evaluated the repeatability of Anterion for measurement of ACD, AL, CCT, and LT, resulting in overall ICC > 0.92 [11]. The same investigative group recorded repeatability for anterior segment measurement, e.g., WTW, angle opening distance (AOD), spur to spur (STS), and trabecular iris space area (TISA), achieving overall ICC > 0.97 [12]. In addition, they reported good repeatability for the whole cornea parameters with ICC > 0.98 [13]. Schiano-Lomoriello et al [14] evaluated repeatability of Anterion, showing ICC > 0.98, with coefficient of variation of CCT, corneal diameter (CD), ACD, LT, and AL < 1%. Intra-device repeatability of Anterion indicates its potential for application in IOL calculation and glaucoma evaluation.

There have been some comparative studies of agreement between IOL Master700 and Anterion. Fisus et al evaluated biometry measurements taken by the devices in 389 eyes of 209 subjects and found that mean absolute difference of keratometry, ACD, LT, AL were significantly different [15]. However, concordance between the devices was high, with differences that were not clinically relevant in IOL calculation. Oh et al [16] reported mean difference of AL 0.005 mm, with ICC 0.999 in their study of 47 eyes of 29 patients. CCT, ACD and keratometry revealed excellent agreement as well. Tana-Rivero et al reported good concurrence of WTW with mean difference of -0.11 mm [17]. Furthermore, Anterion showed good agreement in various biometric parameters with other biometers, e.g. IOL Master500 (a PCI system) [14, 18], Pentacam (Scheimpflug system) [19], MS 39 (combined placido disk and AS-OCT) [14], and CASIA SS 100 (SS-OCT) [20]. Anterion could be applied for reliable irido-corneal angle evaluation prior to and after LPI in PACD patients [21].

PACD and cataracts commonly occur in elderly patients. Thick or anterior position of the lens can affect the iridocorneal angle, leading to acute PAC (APAC) or chronic PACD [22]. PEI appears to be a potential treatment for PACD, achieving good results in IOP control for APAC. Jacobi et al reported that PEI had high success rates in tonometric control in patients who had high ratios of LAF [23]. The EAGLE study demonstrated that clear lens extraction in PACD with initial IOP > 30 mmHg achieved better IOP control than LPI [24]. PEI not only solves glaucoma problems but also improves vision; therefore, accuracy of IOL calculation must be considered in this group of patients.

We applied Barrett formula for IOL calculation in this study. This formula is one of the most accurate and has been widely used, as it has been shown by many studies to achieve better postoperative outcomes than any other formula [25, 26]. The present research demonstrated a significant difference in mean IOL power of 22.30 ± 2.23 D and 22.64 ± 2.34 D from the devices (p< 0.001); however, the arithmetic difference may not have much effect on IOL power selection, since the IOL power is available in 0.5 D increments in every IOL model. We identified that 91.3% of the calculated IOL powers were within ± 0.5 D, and 8.7% were within ± 1 D. As a 1D error in IOL power is equivalent to an error of about 0.67 D in the spectacle plane [27], it appeared to be of no relevance for IOL selection. Prospective evaluation of post-operative IOL accuracy of patients is an ongoing project in our clinic.

Anterior chamber depth is an important parameter for PACD evaluation. Shallow ACD cause irido-trabecular contact, leading to PAS formation and IOP elevation. Marked shallow ACD tends to trigger a severe type of acute attack in PACD. Marchini et al reported ACD, obtained with ultrasonic A-scan, of acute/intermittent PACD, chronic PACD and normal controls as 2.41, 2.77, and 3.3 mm respectively [28]. With OCT systems, our study showed mean ACD of 2.49 and 2.58 mm. IOLMaster700 showed shorter ACD than Anterion, with mean difference of -0.10 mm, and Fisus et al demonstrated a similar result in their cataract patients

[15]. The difference might be related to the algorithm of segmentation of the anterior segment structure: IOLMaster700 determines ACD with the axial distance of anterior cornea to the anterior lens capsule, whereas Anterion determines ACD with CCT plus the aqueous depth (AQD), the axial distance of the posterior cornea to the anterior lens capsule. ACD is not interchangeable between the devices; however, in subgroup analysis, based on ACD at 2.4 mm, the IOL power showed excellent agreement in both subgroups.

LAF is another parameter for prediction of tonometric outcomes of PEI for PACD with higher LAF showing better IOP control. Mean LAF was arithmetically different, at 2.18 vs 2.20 mm, between the devices. However, LAF also demonstrated excellent agreement between them.

## Limitations of the study

The present study had some limitations. First, we did not grade cataracts with Lens Opacity Classification System (LOCS), so we did not know how much the SS-OCT could penetrate various types of cataract. Second, we applied only Barrett universal II formula, which is commonly used in our practice, for IOL calculation, and it was not able to prove the agreement of EPR with the other formulas. The accuracy of IOL calculation should, therefore, be evaluated by post-operative PEI evaluation. Third, because of the growing popularity of toric IOL implantation, the axis of astigmatism from different devices also needs to be evaluated further.

## Conclusion

In conclusion, PACD presenting with small ACD, AL, keratometry and thick lens, was obtained with 2 SS-OCT devices. The biometric measurement performances of the devices were excellent, with low rates of failure. Even though there were some parameters that were arithmetically different between them, they demonstrated excellent agreement of ocular biometry in PACD. Mean difference of AL, a key factor for determination of IOL power, was close to zero. Both devices provided IOL power within an acceptable range, so they can be applied for clinical use with comparable effectiveness.

## Supporting information

**S1 Data. Raw data of the study.**
(XLSX)

## Author Contributions

**Conceptualization:** Boonsong Wanichwecharungruang, Somporn Chantra.

**Data curation:** Boonsong Wanichwecharungruang, Anyarak Amornpetchsathaporn, Wisakorn Wongwijitsook, Somporn Chantra.

**Formal analysis:** Boonsong Wanichwecharungruang, Anyarak Amornpetchsathaporn, Wisakorn Wongwijitsook, Kittipong Kongsomboon, Somporn Chantra.

**Investigation:** Boonsong Wanichwecharungruang.

**Methodology:** Boonsong Wanichwecharungruang.

**Project administration:** Boonsong Wanichwecharungruang.

**Supervision:** Boonsong Wanichwecharungruang, Somporn Chantra.

**Validation:** Boonsong Wanichwecharungruang.

**Visualization:** Boonsong Wanichwecharungruang.

**Writing – original draft:** Boonsong Wanichwecharungruang, Anyarak Amornpetchsatha-porn, Somporn Chantra.

**Writing – review & editing:** Boonsong Wanichwecharungruang, Anyarak Amornpetchsatha-porn, Wisakorn Wongwijitsook, Kittipong Kongsomboon, Somporn Chantra.

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
