## [Decision Letter · Decision Letter 0]

28 Feb 2022

PONE-D-21-35600Evaluation of ocular biometry in primary angle-closure disease with two swept source optical coherence tomography devicesPLOS ONE

Dear Sir,

Thank you for submitting your manuscript to PLOS ONE. After careful consideration, we feel that it has merit but does not fully meet PLOS ONE’s publication criteria as it currently stands. Therefore, we invite you to submit a revised version of the manuscript that addresses the points raised during the review process.

ACADEMIC EDITOR:Based on the reviewer comments attached, we suggest  you to submit the revised version of the manuscript. We encourage you to go through the guidelines for revised manuscript submission to ensure smooth review process. 

We look forward to receiving your revised manuscript.

Kind regards,

Aparna Rao

Academic Editor

PLOS ONE

Journal Requirements:

Reviewers' comments:

Reviewer's Responses to Questions

**Comments to the Author**

1. Is the manuscript technically sound, and do the data support the conclusions?

Reviewer #1: Yes

Reviewer #2: Yes

Reviewer #3: Yes

2. Has the statistical analysis been performed appropriately and rigorously? 

Reviewer #1: Yes

Reviewer #2: Yes

Reviewer #3: Yes

3. Have the authors made all data underlying the findings in their manuscript fully available?

Reviewer #1: Yes

Reviewer #2: Yes

Reviewer #3: Yes

4. Is the manuscript presented in an intelligible fashion and written in standard English?

Reviewer #1: Yes

Reviewer #2: Yes

Reviewer #3: Yes

5. Review Comments to the Author

Reviewer #1: The authors present a nice study about the biometry in eyes with primary angle-closure disease with 2 SST-OCT devices. The study is interesting because one of the device is new and the study population is different to others previously studied.

I only would like to solve some minor issues:

1. Introduction. The authors state that "Primary angle-closure disease (PACD) is the leading cause of blindness worldwide" and cite a paper (reference #1). This paper does not say that. It says that "glaucoma [not PACD] is the leading cause of global irreversible blindness." The authors are mixing reversible and irreversible blindness. And they mix PACD and primary open angle glaucoma (POAG). Please rewrite again the first 4 lines of the introduction.

2. Please write "United States" after every device in which only the state of the USA appears.

3. Please amend "Barrette" and use "Barrett" (lines 245 and 312).

Reviewer #2: This is an interesting study which evaluates ocular biometry in primary angle-closure disease (PACD) with two swept source optical coherence tomography devices. This is probably one of the first study comparing these two systems in PACD eyes. This aspect can be highlighted more.

Reviewer #3: Dr. Somporn Chantra evaluated ocular biometry in primary angle-closure disease with two swept source optical coherence tomography devices. This is meaningful. However, it is advised to calculate the sample size.

6. PLOS authors have the option to publish the peer review history of their article (what does this mean?). If published, this will include your full peer review and any attached files.

Reviewer #1: No

Reviewer #2: No

Reviewer #3: No

---

## [Author Response · Author response to Decision Letter 0]

4 Mar 2022

We have already checked and ensure that our revision manuscript meet the journal requirements (PLOS ONE's style, ORCID ID of corresponding author, supporting information file, and correct references).

5. Review Comments to the Author

Reviewer #1: The authors present a nice study about the biometry in eyes with primary angle-closure disease with 2 SST-OCT devices. The study is interesting because one of the device is new and the study population is different to others previously studied.

I only would like to solve some minor issues:

1. Introduction. The authors state that "Primary angle-closure disease (PACD) is the leading cause of blindness worldwide" and cite a paper (reference #1). This paper does not say that. It says that "glaucoma [not PACD] is the leading cause of global irreversible blindness." The authors are mixing reversible and irreversible blindness. And they mix PACD and primary open angle glaucoma (POAG). Please rewrite again the first 4 lines of the introduction.

Response: Thank you for your comment. We rephrased them as: 

Primary angle-closure glaucoma (PACG) is one of the leading causes of irreversible blindness worldwide. Its global prevalence is showing an upward trend, and it is expected to reach about 66.8 million cases by 2040. Its prevalence is highest in Asia, and people from this region have the greatest number of cases of blindness resulting from glaucoma.1 Primary angle-closure disease (PACD) is a spectrum of this eye condition, including primary angle-closure suspect (PACS), primary angle-closure (PAC) and PACG. (lines 49-54, page 3)

2. Please write "United States" after every device in which only the state of the USA appears.

Response: We added the USA for every device from that country, e.g. IOLMaster700 (Carl Zeiss Meditec, CA, USA), IOL model SN60WF (Alcon, Fort Worth, TX, USA). (line 77, page 4 and line 142, page 7)

3. Please amend "Barrette" and use "Barrett" (lines 245 and 312).

Response: We corrected it as your suggestion. (lines 257 and 325)

Reviewer #2: This is an interesting study which evaluates ocular biometry in primary angle-closure disease (PACD) with two swept source optical coherence tomography devices. This is probably one of the first study comparing these two systems in PACD eyes. This aspect can be highlighted more.

Response: Thank you for your comment.

Reviewer #3: Dr. Somporn Chantra evaluated ocular biometry in primary angle-closure disease with two swept source optical coherence tomography devices. This is meaningful. However, it is advised to calculate the sample size.

Response: We added sample size estimation as:

Sample size estimation

Sample size calculation was performed using the formula for estimating correlation coefficient with a type I error (α) of 0.05 and type II error (β) of 0.2. The authors estimated the correlation of the devices, using the correlation coefficient (r) of 0.7 for the sample size calculation. 

The standard normal deviate for α = Zα = 1.9600

The standard normal deviate for β = Zβ = 0.8416

C = 0.5 * ln[(1+r)/(1-r)] = 0.8673

Total sample size = N = [(Zα + Zβ)/C]2 + 3 = 13

The output indicated that at least 13 subjects should be enrolled. And, in the anticipation of a dropout rate of 30%, the minimum sample size required was 17. (lines 119-128, page 6)

---

## [Editor Report · Decision Letter 1]

9 Mar 2022

Evaluation of ocular biometry in primary angle-closure disease with two swept source optical coherence tomography devices

PONE-D-21-35600R1

Dear Dr. Chantra,

We’re pleased to inform you that your manuscript has been judged scientifically suitable for publication and will be formally accepted for publication once it meets all outstanding technical requirements.

Kind regards,

Aparna Rao

Academic Editor

PLOS ONE
---

## [Editor Report · Acceptance letter]

12 Mar 2022

PONE-D-21-35600R1 

Evaluation of ocular biometry in primary angle-closure disease with two swept source optical coherence tomography devices 

Dear Dr. Chantra:

I'm pleased to inform you that your manuscript has been deemed suitable for publication in PLOS ONE. Congratulations! Your manuscript is now with our production department. 

Kind regards, 

on behalf of

Dr. Aparna Rao 

Academic Editor

PLOS ONE